# Risk Factors Associated with the Development of Metastases in Patients with Gastroenteropancreatic Neuroendocrine Tumors: A Retrospective Analysis

**DOI:** 10.3390/jcm11010060

**Published:** 2021-12-23

**Authors:** Shuzo Kohno, Masahiro Ikegami, Toru Ikegami, Hiroaki Aoki, Masaichi Ogawa, Fumiaki Yano, Ken Eto

**Affiliations:** 1Department of Surgery, The Jikei University Katsushika Medical Center, Tokyo 125-8061, Japan; halm@jikei.ac.jp (H.A.); 0gamasa@jikei.ac.jp (M.O.); 2Department of Pathology, The Jikei University Katsushika Medical Center, Tokyo 125-8061, Japan; ikegami@jikei.ac.jp; 3Department of Surgery, The Jikei University School of Medicine, Tokyo 105-8461, Japan; tikesurg@icloud.com (T.I.); f-yano@live.jp (F.Y.); etoken@jikei.ac.jp (K.E.)

**Keywords:** neuroendocrine tumor, metastasis, lymphatic invasion, venous invasion

## Abstract

Neuroendocrine tumors develop from systemic endocrine and nerve cells, and their occurrence has increased recently. Since these tumors are heterogeneous, pathological classification has been based on the affected organ. In 2019, the World Health Organization introduced a change expected to influence neuroendocrine tumor research, as gastroenteropancreatic neuroendocrine tumors are now included within a unified classification. This retrospective study aimed to investigate the characteristics (e.g., lymph node metastases and all other metastases) of gastroenteropancreatic neuroendocrine tumors using this new classification in 50 cases. Tumor size, depth, MIB-1 index, lymphatic invasion, venous invasion, and neuroendocrine tumor grade were significantly correlated with lymph node metastasis and other metastases. The venous invasion was more strongly correlated with lymph node metastasis and all other types of metastases than with lymphatic invasion. Identification rates for lymphatic invasion were considered lower because of structural problems such as lymphatic vessels being much thinner than veins. However, venous invasion was considered effective in compensating for the low identification rate in cases of lymphatic invasion. In future research, a unified classification and standardized framework for assessment will be important when analyzing the characteristics of neuroendocrine tumors, and large-scale studies are required.

## 1. Introduction

Neuroendocrine tumors (NETs) are heterogeneous malignancies with various pathological and clinical features [1,2,3] that arise from systemic endocrine and nerve cells, and their prevalence has been increasing of late [4]. Previously, NET classification was based on the organ in which the tumor developed. However, according to the 2019 World Health Organization (WHO) classification, NETs occurring in all gastroenteropancreatic (GEP) organs have been grouped and reclassified [5]. Basic and clinical studies have promoted advancements in the diagnosis and treatment of NETs [6,7], and the reclassification of NETs is expected to further advance this research [6].

Treatment guidelines have been created for NETs in each organ, mainly in Europe and the United States [8,9,10,11]. Tumor resection is an effective and important treatment option, with radical treatment involving complete removal of the tumor [8]. Localized NETs are an indication for endoscopic and surgical resection. Indications for endoscopic treatment are determined by the grade, depth, and size of the tumor. For localized tumors that are not indicated for endoscopic resection, the indication for surgical resection is determined based on whether the metastatic lesion can be completely resected [8,12,13]. Thus, the diagnosis and prediction of metastasis, especially lymph node metastasis, is important when selecting the treatment option for a particular NET. Lymph node metastases may be surgically removed, which can improve prognosis; however, it is difficult to surgically remove other metastases, often not indicated for surgery, thus highlighting the need for preoperative evaluation.

Therefore, in the present study, we aimed to examine the factors related to lymph node metastases and all other types of metastases in the new GEP-NET classification and determine problems that can arise when identifying these factors.

## 2. Materials and Methods

### 2.1. Patients

The medical histories of 48 patients with 50 consecutive cases of GEP-NET treated via endoscopic or surgical resection at our institution between January 2010 and March 2021 were retrospectively collected and compared. Patients who refused treatment and cases in which the tumor size was unknown were excluded. Data related to age, sex, body mass index, and pathological tumor findings (site, size, depth of invasion, lymphatic invasion, venous invasion, and MIB-1 index) were obtained from electronic medical records for all patients.

### 2.2. Pathological Classification and Staging

The diagnosis and treatment of patients were evaluated based on contemporary standards. The NET classification was based on the 2019 World Health Organization classification [14]. All GEP-NETs were classified into well-differentiated NETs, poorly differentiated neuroendocrine carcinomas, and mixed endocrine/non-endocrine neoplasms. Well-differentiated NETs were classified into grades 1, 2, and 3 (G1, G2, and G3) based on the mitotic rate and Ki-67 index (G1, mitotic rate of <2 per 10 high-power fields and/or Ki67 index of <3%; G2, mitotic rate of 2 to 20 per 10 high-power fields and/or Ki67 index of 3 to 20%; and G3, mitotic rate of >20 per 10 high-power fields and/or Ki67 index of >20%). Neuroendocrine carcinomas were classified as small- or large-cell types. Mixed endocrine/non-endocrine neoplasms consisted of either neuroendocrine or non-neuroendocrine components, such as adenocarcinoma. For endoscopically resected specimens, a cut surface was created at the center of the lesion. In addition to hematoxylin and eosin staining, synaptophysin, chromogranin, CD56, and Ki67 staining were performed for all resected lesions. The lymphatic invasion was diagnosed via D2-40 staining [15], and venous invasion was diagnosed via Elastica van Gieson staining [16].

### 2.3. Statistical Analysis

Univariate and multivariate analyses for age, sex, body mass index, tumor size, site of origin, depth, MIB-1 index, lymphatic invasion, venous invasion, and NET grade for lymph node metastases and all metastases were performed using logistic regression analysis. The effects of lymphatic invasion and venous invasion on lymph node metastases and all metastases were analyzed using Fisher’s exact test. Statistical significance was set at *p* < 0.05. Statistical analyses were performed using SPSS Statistics version 22.0 (IBM Japan, Ltd., Tokyo, Japan).

## 3. Results

### 3.1. Background Data

The background data of all patients with GEP-NET who underwent endoscopic or surgical resection at our institution during the study period are shown in Table 1. Fifty resections were performed in 48 patients with NETs. One case had multiple NET occurrences in the duodenum, ileum and rectum, and the other 49 cases had a NET from a single site.

Three deaths were noted in our study. The first patient died due to brain metastasis of esophageal neuroendocrine carcinoma three years and ten months after a subtotal esophagectomy; the second patient died of sepsis due to a perianal abscess with rectal, duodenal, and intraperitoneal recurrence seven years and five months after endoscopic resection for rectal NET G2; and the third patient died two months after sigmoid resection for a perforated intraperitoneal abscess of a sigmoid colon NET with liver and multiple lymph node metastases.

The average observation period was 925.3 (36~3000) days. There were 35 (70%) men and 15 (30%) women, and the patient age ranged from 33–88 years (average, 60.4 years). The mean body mass index was 23.67 ± 4.11. The resection method was endoscopic resection in 36 cases and surgical resection in 14 cases. Surgical resection was considered in cases performed after endoscopic resection and those performed after examination without endoscopic resection. The esophagus was resected in two cases, the stomach in six, the duodenum in seven, the small intestine in two, the pancreas in three, the colon in two, and rectum in 31. The invasion depth was intramucosal in seven cases, submucosal (in the pancreas, the tumor remained in the organ and did not infiltrate adjacent organs) in 34 cases, up to the muscularis propria in six cases, up to the serosa in two cases, and extraserosal in one case. Metastasis was observed in eight cases (16.7%). The metastatic sites were observed in the lymph nodes, liver, lung, and brain in six, three, one, and one case, respectively, and there was one case of dissemination.

### 3.2. Venous and Lymphatic Invasion

Lymphatic invasion of resected specimens was reported in eight cases before this study, but two small metastatic lesions (Figure 1) were diagnosed again by a different pathologist, indicating a total of 10 cases (20%). The lesions identified via repeat microscopy with D2-40 staining exhibited very small amounts of NET cells in vertically or diagonally cut lymphatic vessels. Intravenous invasion was observed in 13 cases (26%). Venous invasion was identified in all six cases of lymph node metastasis, while lymphatic invasion was identified in four cases.

The univariate analysis revealed significant differences in tumor size (Odds ratio [OR] = 1.033, 95% confidence interval [CI]: 1.001–1.066, *p* = 0.044), depth (OR = 3.957, 95% CI: 1.306–11.992, *p* = 0.015), MIB-1 index (OR =6.800, 95% CI: 1.082–42.731, *p* = 0.041), lymphatic invasion (OR = 12.677, 95% CI: 1.888–84.965, *p* = 0.009), and NET grade (OR = 1.724, 95% CI: 1.874–24.131, *p* = 0.003) in cases of lymph node metastases, although statistical values could not be obtained for venous invasion (Table 2). Similarly, for all other metastases, the univariate analysis identified significant differences in tumor size (OR = 1.097, 95% CI: 1.020–1.180, *p* = 0.013), depth (OR = 9.253, 95% CI: 2.038–42.013, *p* = 0.004), MIB-1 index (OR = 6.111, 95% CI: 1.222–30.572, *p* = 0.028), lymphatic invasion (OR = 6.000, 95% CI: 1.172–30.725, *p* = 0.032), venous invasion (OR = 35.000, 95% CI: 3.700–331.059, *p* = 0.002), and NET grade (OD = 14.900, 95% CI: 2.979–74.529, *p* = 0.001) (Table 3). However, only NET grade exhibited a significant difference in the multivariate analysis. In the multivariate analysis, we used the variable increase method, and each variable was included in the multivariate model based on the likelihood ratio. Therefore, the OR was only calculated for the NET grade.

### 3.3. Correlations with Lymph Node Metastasis and All Other Types of Metastases

We investigated the relationship between lymph node invasion and venous invasion in cases of lymph node metastasis (Table 4) and all other types of metastases (Table 5). For lymph node metastases, the *p*-value for lymphatic invasion was 0.011, whereas that for venous invasion was 0.000, indicating a stronger correlation with venous invasion. Furthermore, for all metastases, the *p*-value for lymphatic invasion was 0.041, whereas that for venous invasion was 0.000, indicating a stronger correlation with venous invasion.

## 4. Discussion

The curative treatment for GEP-NET is complete resection. However, given the heterogeneous nature of the lesions, indications for resection have been examined based on the organ affected. We believe that it is necessary to examine the entire NET to define its characteristics. In this study, we investigated the factors that influence lymph node metastasis and other types of metastases in patients with GEP-NET. Our analysis revealed that both categories of metastasis were significantly associated with tumor size, depth, MIB-1 index, lymphatic invasion, venous invasion, and NET grade. This result is consistent with currently reported organ-specific results [10].

The prognosis of lymph node metastasis is important for treating NETs [13,17]. A localized NET indicates endoscopic or surgical resection; however, surgical resection requires complete resection of lymph node metastases; therefore, optimal methods for lymph node dissection are being investigated [9,11,18]. NETs in the stomach, duodenum, pancreas, colon, and rectum that are <1 cm in size and intramucosal ly0 and v0 tumors are indicated for endoscopic treatment [19,20,21,22,23,24]. There is no treatment algorithm for esophageal NETs due to the scarcity of cases [25]. Endoscopic resection is not indicated for NETs in the small intestine—even if the tumor’s major axis is 1 cm or less—due to the high rate of lymph node metastasis [26], a large number of multiple lesions [27], and technical difficulty. The amount of tumor remaining in the small intestine after resection may be larger than the piece resected; therefore, it may be necessary to consider the depth and size of the whole tumor when analyzing the data. In pancreatic NET staging, the criteria for invasion depth differ from the diagnostic criteria in other areas of the gastrointestinal tract, making them difficult to evaluate using the same criteria. For these reasons, when considering treatment options for a GEP-NET as a whole, it is necessary to consider the organ in which the tumor is situated.

Our findings indicated that venous invasion was more strongly correlated with lymph node metastases and all other metastases than lymphatic invasion. Previous studies have demonstrated that immunostaining increases the detection rate of vascular invasion [16,28]. However, other studies have reported contradictory data regarding the pathological identification of vascular invasion with and without immunostaining, and this discrepancy must be fully considered when comparing the data reported. Overexpression of a large number of angiogenesis-promoting molecules has been reported in NET cells, suggesting a link to metastasis [29]. A study on early-stage colorectal cancer noted that venous invasion was more useful than lymphatic invasion as a predictor of lymph node metastasis [30]. For small rectal NETs of 1.5 cm or less, those with vascular infiltration have been reported to have a high potential for lymph node metastasis, as high as 48.8% [31], but the authors did not compare lymphatic infiltration and venous invasion. Most studies have examined the possibility of metastasis in cases of vascular invasion, which is a combination of lymphatic invasion and venous invasion, and few have compared lymphatic invasion with venous invasion. In venous invasion, a large number of tumor cells are identified in the lumen, and the number is greater than that observed in lymphatic vessels. In lymphatic invasion, a few tumor cells are often found in the smaller lymphatic lumen. The difficulty in identifying lymphatic invasion, especially when the lymphatic vessels are cut vertically, may be related to the low rate at which lymphatic invasion is identified. These factors may explain the higher identification rate for venous invasion than for lymphatic invasion. Improving the rate at which lymphatic invasion is identified requires careful tumor identification using D2-40 immunostaining for all diagnoses. Combined evaluation with synaptophysin and other immunostaining methods can also be effective. In this study, venous invasion was observed in all cases of lymph node metastasis, and venous invasion correlates well with lymph node metastasis at present; therefore, simultaneous evaluation of vascular invasion is considered a sufficient index.

A previous study reported that lymphatic invasion and lymph node metastasis rates increase significantly in cases of multiple lesions in patients with rectal NETs [32]. Furthermore, lymph node metastasis of rectal NETs has been associated with tumor size, depth of invasion, vascular invasion, and WHO grade, which is consistent with our findings for GEP-NET [33]. In addition, lymph node metastasis has been reported to affect prognosis. In patients with gastric NETs undergoing gastrectomy and lymph node dissection, the prognosis is related to type I and type III of the Rindi classification, tumor size, and grade [34]. For gastric NETs, the Rindi classification is based on the presence or absence of atrophic gastritis and gastrin secretion. A treatment algorithm has been created based on this classification [23]. Comparing findings for tumors based on this classification and those based on the entire NET may provide insight into methods for improving NET classification. Previous research has indicated that venous invasion is a poor prognostic indicator in patients with pancreatic NETs [35]. In the current study, results for GEP-NET metastasis were similar to those for individual organs. In particular, the results were quite similar to those for the rectum, although most of our data were from rectal cases. Although differences in organ specificity may influence GEP-NET research, we believe that these findings are important for the study of tumor specificity and reflect the GEP-NET group.

The present study had some limitations, including its retrospective design and the low number of cases for certain organs. Further, lymph node dissection was not performed in all cases, suggesting that the rate of metastasis was not accurate in the lymph node metastasis group and that standardized and appropriate diagnostic imaging methods for lymph node metastasis remain unclear. Additionally, follow-up for recurrence may have been insufficient, and the overall assessment may have included unsuitable organs. Randomized controlled studies are difficult, given that the guidelines indicate appropriate treatments. Further studies are warranted to obtain necessary and sufficient data based on very accurate common diagnostic criteria. These studies should employ appropriate follow-up periods and a thorough, unified method for investigating recurrence.

## 5. Conclusions

This study investigated factors related to the metastasis of GEP-NETs based on 2019 WHO classification. The most important factors were found to be tumor grade and vascular invasion. Of the types of vascular invasion, venous invasion was more highly correlated with metastasis than lymphatic invasion, and the analysis indicated that pathological examination of lymphatic invasion might be problematic. At present, it may be better to evaluate vascular invasion to assess NET metastasis, as it combines lymphatic and venous invasion. The development of a unified classification system for NETs and a standardized method for evaluating them is important for the future of NET research.

## Figures and Tables

**Figure 1 jcm-11-00060-f001:**
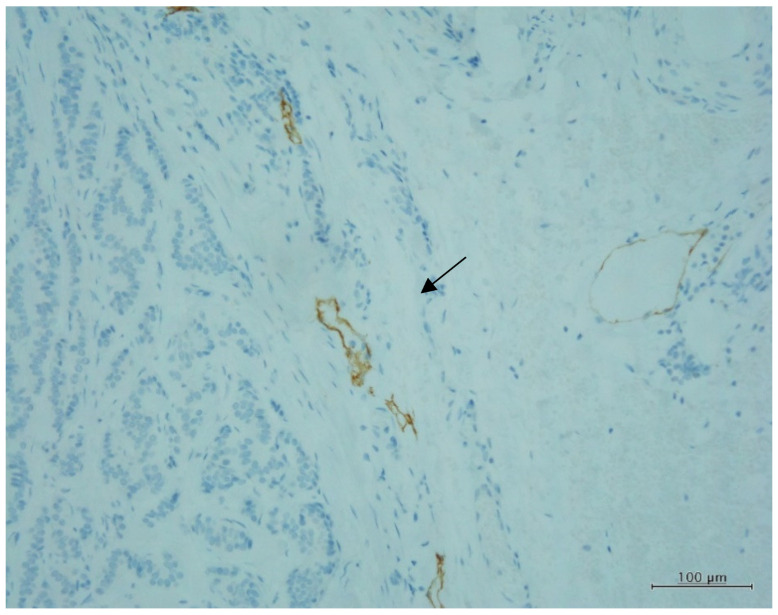
Rectal G1 NET 6 mm. D2-40 immunohistochemical stain showing lymphatic invasion (arrow) in the lymphatic vessels that had been cut diagonally. (×100).

**Table 1 jcm-11-00060-t001:** Background characteristics of neuroendocrine tumors (*n* = 50).

Characteristic		Value
Age (years)	Mean ± SD	60.3 ± 14.1
Sex, *n* (%)	Male (%)	35 (70)
	Female (%)	15 (30)
Body mass index	Mean ± SD	23.67 ± 4.11
Size (mm)	Mean ± SD	9.06 ± 9.22
Excision method	Endoscopic resection	36 (72%)
	Surgical resection	14 (28%)
Location	Esophagus	2 (3.8%)
	Stomach	6 (11.5%)
	Duodenum	7 (13.5%)
	Small intestine	2 (3.8%)
	Pancreas	3 (5.8%)
	Colon	1 (1.9%)
	Rectum	31 (59.6%)
Depth	T1a	7 (14%)
	T1b(+c)	34 (68%)
	T2	6 (12%)
	T3	2 (4%)
	T4	1 (2%)
MIB-1 index	<3%	36 (72%)
	>3%	14 (28%)
Grade	Grade 1	39 (78%)
	Grade 2	7 (14%)
	Grade 3	0
	NEC	4 (8%)
Characteristic	Yes	No
Lymphatic invasion (*n*)	10 (20%)	40 (80%)
Venous invasion (*n*)	13 (26%)	37 (74%)

SD, standard deviation; NEC, neuroendocrine carcinoma. T1a, intramucosal; T1b, submucosal; T2, muscularis propria; T3, subserosal; T4, extraserosal infiltration. In the pancreas, T1: localized to the pancreas (maximum diameter ≤2 cm), T2 (localized to the pancreas, 2 cm < maximum diameter ≤ 4 cm), T3: (localized to the pancreas, 4 cm < maximum diameter/duodenum/bile duct infiltration).

**Table 2 jcm-11-00060-t002:** Logistic regression analysis for lymph node metastases.

	Univariate Analysis					Multivariate Analysis			
	OR	95% CI			*p*-Value	OR	95% CI		*p*-Value
Age (per 1 year)	1.074	0.988		1.166	0.092	–			
Sex, Male (vs. Female)	0.375	0.066		2.120	0.267	–			
Body mass index (per 1 kg/m^2^)	0.866	0.682		1.100	0.239	–			
Size (per 1 mm)	1.033	1.001		1.066	0.044	n.e.			
Location						–			
Rectum	1.000		ref						
Stomach	n.c.								
Duodenum	6.000	0.321		112.258	0.231				
Esophagus	n.c.								
Intestine	n.c.								
Large intestine	n.c.								
Pancreas	n.c.								
Depth (per 1)	3.957	1.306		11.992	0.015	n.e.			
MIB-1 index, >3% (vs. <3%)	6.800	1.082		42.731	0.041	n.e.			
Lymphatic invasion, Yes (vs. No)	12.667	1.888		84.965	0.009	n.e.			
Venous invasion, Yes (vs. No)	n.c.					–			
Lymphatic invasion/Venous invasion						–			
Neither	1.000		ref						
Either one/Both	n.c.								
Grade (per 1)	6.724	1.874		24.131	0.003	6.724	1.874	24.131	0.003

OR, odds ratio; 95% CI, 95% confidence interval; ref, reference; n.c., not calculable; n.e., not entered. Variables that were significant in the univariate analysis were used in the multivariate analysis (variable increase method: likelihood ratio). Values in bold indicate significant factors.

**Table 3 jcm-11-00060-t003:** Logistic regression analysis for all metastases.

	Univariate Analysis					Multivariate Analysis			
	OR	95% CI			*p*-Value	OR	95% CI		*p*-Value
Age (per 1 year)	1.066	0.993		1.145	0.077	–			
Sex, Male (vs. Female)	0.355	0.076		1.667	0.189	–			
Body mass index (per 1 kg/m^2^)	0.901	0.737		1.102	0.309	–			
Size (per 1 mm)	1.097	1.020		1.180	0.013	n.e.			
Tumor location						–			
Rectum	1.000		ref						
Stomach	n.c.								
Duodenum	2.900	0.219		38.320	0.419				
Esophagus	n.c.								
Intestine	n.c.								
Large intestine	n.c.								
Pancreas	7.250	0.443		118.700	0.165				
Depth (per 1)	9.253	2.038		42.013	0.004	n.e.			
MIB-1 index, >3% (vs. <3%)	6.111	1.222		30.572	0.028	n.e.			
Lymphatic invasion, Yes (vs. No)	6.000	1.172		30.725	0.032	n.e.			
Venous invasion, Yes (vs. No)	35.000	3.700		331.059	0.002	n.e.			
Lymphatic invasion/Venous invasion						n.e.			
Neither	1.000		ref						
Either one/both	17.500	1.940		157.881	0.011				
Grade (per 1)	14.900	2.979		74.529	0.001	14.900	2.979	74.529	0.001

OR, odds ratio; 95% CI, 95% confidence interval; ref, reference; n.c., not calculable; n.e., not entered. Variables that were significant in the univariate analysis were used in the multivariate analysis (variable increase method: likelihood ratio). Values in bold indicate significant factors.

**Table 4 jcm-11-00060-t004:** Comparison of cases with and without lymph node metastases.

	Without Lymph Node Metastases		With Lymph Node Metastases		*p*-Value
	(*n* = 44)		(*n* = 6)		
Lymphatic invasion					0.011
No	38	86.4	2	33.3	
Yes	6	13.6	4	66.7	
Venous invasion					0.000
No	36	81.8	0	0.0	
Yes	8	18.2	6	100.0	
Lymphatic invasion/Venous invasion					0.000
Neither	31	70.5	0	0.0	
Either one	12	27.3	2	33.3	
Both	1	2.3	4	66.7	
Lymphatic invasion/Venous invasion					0.002
Neither	31	70.5	0	0.0	
Either one/both	13	29.5	6	100.0	

Data are presented as *n* %; *p*-value: Fisher’s exact test. Values in bold are significant.

**Table 5 jcm-11-00060-t005:** Comparison of cases with and without all metastases.

	Without All Metastases		With All Metastases		*p*-Value
	(*n* = 42)		(*n* = 8)		
Lymphatic invasion					0.041
No	36	85.7	4	50.0	
Yes	6	14.3	4	50.0	
Venous invasion					0.000
No	35	83.3	1	12.5	
Yes	7	16.7	7	87.5	
Lymphatic invasion /Venous invasion					0.000
Neither	30	71.4	1	12.5	
Either one	11	26.2	3	37.5	
Both	1	2.4	4	50.0	
Lymphatic invasion /Venous invasion					0.003
Neither	30	71.4	1	12.5	
Either one/both	12	28.6	7	87.5	

Data are presented as *n* %; *p*-value: Fisher’s exact test. Values in bold are significant.

## Data Availability

The data presented in this study are available in this article.

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
