# Peer review of "Risk Factors Associated with the Development of Metastases in Patients with Gastroenteropancreatic Neuroendocrine Tumors: A Retrospective Analysis"

_jcm, 2021, doi:10.3390/jcm11010060_

Round 1

Reviewer 1 Report

The statistical analys is correct. It could be improved by increasing the number of patients

Author Response

Thank you very much for reviewing our manuscript and offering valuable advice.

We have addressed your comments with point-by-point responses and revised the manuscript accordingly.

Your opinion regarding increasing the number of cases is reasonable; however, we had applied for and registered for 50 cases in this study. Hence, it will take time to re-apply to the Institutional Review Board to increase the number of cases. Therefore, we are unable to increase the sample size. My apologies!

Reviewer 2 Report

This manuscript investigates the well know prognostic risk factors of gastroenteropancreatic neuroendocrine tumours, with a focus on vascular invasion. The two main types of vascular invasion (venous and lymphovascular) are handled separately using elastica van Gieson staining and D2-40 immunostaining. The specimens investigated were obtained by either surgical or endoscopic tumour resections. The outcome measures were the development of metastases, either in regional lymph nodes (6 cases) or other distant organs (4 cases). The authors properly highlight the main limitations of this retrospective study (low number of patients and tumours). Another important limitation is the relatively short follow-up time. 

Besides confirming the importance of other well-known risk factors, the authors found that both types of vascular invasion have a prognostic impact on these tumours. 

Specific comments and suggestions: 

  1. Although the number of examined tumours is 50 throughout the text, the location section of Table 1 shows altogether 52 primaries.
  2. The average observation period was 925 days. The authors should give also the min-max values. 
  3. A short follow-up time was an exclusion criterium in this study?
  4. It is the impression of the reviewer, that the authors do not use consequently the terms "all metastases" and "all other metastases" (see titles of Tables 3 and 5, and title of paragraph 3.3.

Author Response

Thank you very much for reviewing our manuscript and offering valuable advice.

We have provided point-by-point responses to your comments and have revised the manuscript accordingly.

  1. Your opinion is reasonable. Although the number of examined tumours is 50 throughout the text, the location section of Table 1 shows altogether 52 primaries.
  2. The average observation period was 925 days. The authors should give also the min-max values. 
  3. A short follow-up time was an exclusion criterium in this study?
  4. It is the impression of the reviewer, that the authors do not use consequently the terms "all metastases" and "all other metastases" (see titles of Tables 3 and 5, and title of paragraph 3.3.

  1. A total of 52 sites in 50 patients; 1 patient had NET in 3 sites, and the lesion in the rectum was the most advanced; hence, it was considered as rectal NET. I have added the following information under 3.1. Background. “One patient had multiple NET lesions in the duodenum, ileum and rectum, and the other 49 patients had NET in a single site.”
  2. I have inserted the observation period.
  3. Patients with a short follow-up time, in whom the existence of metastasis was examined, were also included in the study analysis.
  4. As you pointed out, we have unified it as “all metastases."

Thank you for your kind guidance.

Reviewer 3 Report

The Study by Kohno et al. investigated factors related to the metastasis of GEP-NETs based on the 2019 WHO classification. The most important factors were found to be tumor grade and vascular invasion. It is an interesting study that should be employed on a larger cohort with appropriate follow-up periods. Despite the small and heterogenous sample size which is actually reflecting the rare character of NET.

 Here are some recommendations:

  1. The authors should provide data in percent for all parameters in table 1 not only gender.

  1. The authors should provide information in the figure legends about the tumor origin of the patient shown in figure 1.

  1. There are several grammer and language mistakes throughout the manuscript. These should be corrected during major revision.

Author Response

Thank you very much for reviewing our manuscript and offering valuable advice.

We have provided point-by-point responses to your comments and have revised the manuscript accordingly.

  1. The authors should provide data in percent for all parameters in table 1 not only gender.
  1. The authors should provide information in the figure legends about the tumor origin of the patient shown in figure 1. 
  1. There are several grammer and language mistakes throughout the manuscript. These should be corrected during major revision.

  1. Thank you for your advice. I have added ‘’%’’ sign to the data.
  2. As you have pointed out, the site of occurrence should be presented. I have included it in the legend of figure 1
  3. The manuscript was initially proofread by a professional Editing company (Editage). Accordingly, I have consulted them again for another round of proofreading.

Thank you for your kind guidance.

This manuscript is a resubmission of an earlier submission. The following is a list of the peer review reports and author responses from that submission.